# Patient Perceptions of Paramedian Minimally Invasive Spine Skin Incisions

**DOI:** 10.3390/jpm13060878

**Published:** 2023-05-23

**Authors:** Kimberly Quiring, Morgan P. Lorio, Jorge Felipe Ramírez León, Paulo Sérgio Teixeira de Carvalho, Rossano Kepler Alvim Fiorelli, Kai-Uwe Lewandrowski

**Affiliations:** 13700 E. Williams Field Rd., Gilbert, AZ 85295, USA; butterworthpa@gmail.com; 2Advanced Orthopedics, 499 E. Central Pkwy, Ste. 130, Altamonte Springs, FL 32701, USA; mloriomd@gmail.com; 3Minimally Invasive Spine Center, Reina Sofía Clinic, Bogotá 104-76, Colombia; jframirezl@yahoo.com; 4Department of Orthopaedics, Fundación Universitaria Sanitas, Bogotá 104-76, Colombia; 5Pain and Spine Minimally Invasive Surgery Service, Gaffre e Guinle University Hospital, Rio de Janeiro 20270-004, RJ, Brazil; paulo.carvalho@unirio.br; 6Department of General and Specialized Surgery, Gaffrée e Guinle University Hospital, Federal University of the State of Rio de Janeiro (UNIRIO), Rio de Janeiro 20270-004, RJ, Brazil; fiorellirossano@hotmail.com; 7Center for Advanced Spine Care of Southern Arizona, Tucson, AZ 85712, USA; 8Department of Orthopedics, Hospital Universitário Gaffre e Guinle, Universidade Federal do Estado do Rio de Janeiro, Rio de Janeiro 20270-004, RJ, Brazil

**Keywords:** novel mini oblique, larger intersecting, vertical, horizontal, paramedian minimally invasive lumbar spine skin incisions, patient satisfaction

## Abstract

Background: In clinical outcome studies, patient input into the factors that drive higher satisfaction with lumbar minimally invasive spinal surgery (MISS) is rare. The skin incision is often the only visible consequence of surgery that patients can assess. The authors were interested in patients’ opinions about the type of lumbar paramedian minimally invasive spinal (MIS) skin incision employed during MISS and how novel skin incisions could impact patients’ interpretation of the outcome. The authors wanted to compare traditional lumbar stab incisions to three novel lumbar paramedian (MIS) skin incisions to determine if further study is indicated. The primary objective was to examine patient satisfaction and perceptions regarding lumbar paramedian MIS skin incisions. Methods: We reviewed the literature and conducted a patient opinion survey. Responses were solicited from back pain patients from a single chiropractic office. Survey questions regarding novel skin incisions for minimally invasive spine surgery (NSIMISS) were conceptualized. The three novel skin incisions were designed using Langer’s lines to reduce the total number of incisions; improve patient satisfaction; increase ease of surgical approach/fixation; and reduce operative time/radiation exposure. Results: One hundred and six participants were surveyed. When shown traditional lumbar paramedian MIS skin stab incisions, 76% of respondents indicated negative responses, *n* = 65. The majority of patients chose traditional stab incisions (*n* = 41) followed by novel larger intersecting incisions (*n* = 37). The least popular incisions were the novel horizontal (*n* = 20) and the novel mini oblique (*n* = 5) incisions. Female patients worried more than male patients about how their incision looked. However, there was no statistically significant difference (*p* value of 0.0418 via Mann–Whitney U one-tailed test and *p* value of 0.0836 via Mann–Whitney U two–tailed test). Patients less than or equal to 50 years of age worried more than patients over 51 years of age, which was statistically significant (*p* value of 0.0104 via Mann–Whitney U one-tailed test and *p* value of 0.0208 via Mann–Whitney U two-tailed test). Conclusions: Patients do have opinions on the type of lumbar paramedian MIS skin incision used. It appears that younger patients and female patients worry most about how the incision on their back looks after surgery. A larger population of patients across many demographics is needed to validate these findings.

## 1. Introduction

Factors impacting patients’ opinions regarding their lumbar skin incisions range from the type of surgery to the location and size of the incision. While recovery and skin incision healing are uneventful in most patients, wound complications and infections may lead to delayed healing or disfiguring scars. Such problems could prolong recovery and require additional intervention and revision surgeries. In addition, patients’ perceptions of their lumbar spine surgery incisions may be influenced by their perception of the overall clinical outcome, in the context of their biased expectations and the quality of care and support they receive from their healthcare providers.

In this study, the authors stipulated that the impact of lumbar spine incisions on patient satisfaction with clinical outcomes after spine surgery likely depends on several factors, including the type of surgical procedure performed, the extent of the incision, and the patient’s overall health status. The contemporary literature suggests that minimally invasive procedures requiring smaller incisions result in higher patient satisfaction rates than traditional open surgery, which mandates larger incisions [1,2,3,4]. Minimally invasive procedures can result in less blood loss, a shorter hospital stay, and a faster recovery time, which may contribute to higher patient satisfaction [1]. However, not all patients are candidates for minimally invasive surgery, and some conditions may require a larger incision. In these cases, patient education and communication can be essential in managing expectations and addressing any concerns related to the incision size.

There is little evidence in the published peer-reviewed literature that allows isolating the impact of the lumbar skin incisions on patient satisfaction from other commonly employed clinical outcome measures after spine surgery, including the following: the level of pain relief; the ability to return to work and normal activities; and the overall quality-of-life improvement. Further, the authors of this article could not find any evidence in the literature comparing patient satisfaction driven by skin incision perception to effective pain management, postoperative rehabilitation, and ongoing follow-up care, not discounting the wide range of factors beyond the surgical procedure itself, including the patient’s preexisting health conditions, expectations, and overall experience with the healthcare system, that may impact patient satisfaction with lumbar spine surgery [5,6]. Better understanding of what goes into patient satisfaction is critical to successful spine care delivery as many payers implement payment models considering the satisfaction of their beneficiaries with the provided service. Spine surgeons will likely have to better communicate the expected outcomes and recovery process for their specific surgery. Additionally, spine surgeons will need to provide enough information about the postoperative course and desired outcome in the future so that patients feel included in the decision-making process regarding whether to undergo these often elective lumbar spine operations [7]. 

Despite these apparent limitations imposed by the multitude of confounding factors, the authors of this prospective survey study were interested in soliciting responses from patients regarding their lumbar spine skin incisions and how these responses could impact a patient’s perception of the clinical outcome. Patient input into the factors that drive higher satisfaction with minimally invasive spinal surgery (MISS) is rare. There is little research on patient input or satisfaction regarding lumbar paramedian minimally invasive spinal (MIS) skin incisions and post-surgical scars and their significance to patients [8]. While the advantages and disadvantages of minimally invasive surgery are well known, the impact of the cosmetic appearance on patient satisfaction is poorly understood. However, the skin incision is often the only visible consequence of spine surgery for patients. Higher patient satisfaction scores are currently impacting reimbursements to physicians and facilities [9,10]. Langer’s lines run obliquely in the thoracic spine [11]. If an oblique incision were used, it could result in a better cosmetic result. In the lumbar spine, a horizontal incision may be advantageous. Novel mini oblique lumbar incisions may allow for easier rod insertion in patients with scoliosis, as accomplishing this through misaligned stab incisions is problematic. 

The authors of this study were interested in patients’ opinions about the type of skin incision employed during MISS and how novel skin incisions could impact patients’ interpretation of the outcome. In addition, the authors wanted to know if there is a role for a novel type of incision in MIS surgery and how these novel incisions would be perceived by patients compared to traditional stab incisions. In a paper survey, they solicited responses from patients who have back pain seeking care at a single rural chiropractic office in Tennessee to find out, when given a choice between incisions for a multilevel fusion, which incision would most patients choose and for what reason.

## 2. Materials and Methods

### 2.1. Study Design

Patients were enrolled into this prospective survey study between February 2022 and March 2022. There were 53 female patients and 50 male patients. The average age was 51 years. 

### 2.2. Survey Methods

A chiropractic office was used to facilitate surveys in a rural area in Tennessee, where not everyone pays for, can afford, or even has access to the internet. In a rural chiropractic office, there is a high daily census of back pain patients with very few confounding variables, such as long wait times and secondary gain. The authors desired to complete a larger study using the Typeform^TM^ platform, expanding the recruitment for this study. They were limited by restrictions unfortunately imposed by the local Institutional Review Board (IRB), which would not approve the use of the Typeform^TM^ platform. The local IRB claimed that protection of the data could not be verified and solicitation of subjects would violate their own specific protocols. Advantages of the Typeform^TM^ are included in the discussion. 

### 2.3. Incision Research

The first author provided a paper survey to 106 back pain patients undergoing conservative chiropractic therapy. After informed consent was obtained, patients hand completed paper form surveys. One hundred and three patients successfully completed the paper survey in a sufficient way to accomplish and complete data analysis. Three patients were excluded for incomplete surveys. 

In the survey, responses were solicited either with a Likert scale from 1–5 or from a list of multiple choice answers. Direct written text responses were also possible. Patients were asked the following questions:(1)Can you describe your first impression when you see the incisions that could be on your back after spine surgery? (As shown in Figure 1 a comment section was provided.)(2)How much do you worry about how the incision on your back looks? (Likert scale from 1–5 stars; ✰✰✰✰✰).(3)Which of the following comes to mind if you had a choice of a skin incision on your back?a.I prefer a midline skin incision;b.I want the least number of skin incisions possible;c.I am more worried that I have less pain;d.I am more concerned with my surgeon being able to do a good job.(4)Please rate how this incision appeals to you? Figure 2a.(5)Please rate how this incision appeals to you? Figure 2d.(6)Please rate how this incision appeals to you? Figure 2e.(7)Please rate how this incision appeals to you? Figure 2b.(8)Which of the four incisions would you prefer? Figure 2a,b,d,e.(9)Please tell us anything about your incision from spine surgery!

The authors were blinded as to the responding patients’ identities. Upon termination of the survey, the responses were entered in an Excel file format and imported into an IBM SPSS (version 27) statistical software package for further data analysis and reported to the local IRB. Patient surveys and consent forms were disposed of utilizing a local IRB approved shred box at a local hospital.

### 2.4. Statistical Analysis

Descriptive statistical measures were used to count responses and calculate the mean, range, standard deviation, and percentages. Chi-square statistics were used to determine the strength of the association between responses. Wherever applicable, a *p* value of 0.05 or less was considered statistically significant, and a confidence interval of 95% was employed for all statistical tests. For question number one, “Can you describe your first impression when you see the incisions that could be on your back after spine surgery?” as shown in Figure 1, responses were reviewed by the first author. Written responses were divided into two groups by marking them as positive or negative. Missing and neutral responses were not studied, *n* = 18. Typical positive responses included “small and minimal”, “acceptable”, and minimally invasive”. Typical negative responses included “ouch”, “awful”, and “this looks painful to look at”. Once the two groups were identified, the responses were counted. Finally, the number of responses in each group was divided by the total survey responses and then multiplied by 100 to get the percentage. Question two was recorded in an Excel spread sheet with the corresponding age and sex. For comparison, four groups were created and compared. Group one consisted of those up to the age of 50, as compared to group two, which consisted of those aged 51 and older. Group three consisted of females, as compared to group four, which consisted of males. The Likert response from each survey was entered into IBM SPSS software and the Mann–Whitney U test was calculated using one- and two-tailed parameters. The *p* values from all four tests were reviewed, with a *p* value of less than 0.05 regarded as statistically significant. Responses from questions three and nine were not studied. Question three had too few responses to compare, as the results divided the total responses of 103 into four subsets of smaller size. A larger sample size for each result is required to determine the statistical significance of question three. Questions four through eight were counted and divided by the total survey responses and then multiplied by 100 to get the percentage. Question nine was not studied since only six patients wrote in this section, and it did not relate to the primary or secondary hypothesis. 

## 3. Results

In our prospective study, 106 back pain patients were surveyed. Three patients were excluded for not completing all survey questions. Hence, the response rate was 97.17%. Analysis was performed focused on demographics of age, sex, and opinion related to the primary hypothesis of “Do back pain patients have opinions on the type of paramedian lumbar MIS skin incision used?” Our secondary hypothesis was “Is there a role for novel MIS skin incisions and how do these rank among patients?” Patients were shown an image of intraoperative multilevel stab incisions (Figure 1) and asked, “Can you describe your first impression when you see these incisions that could be on your back after spine surgery”? Seventy-six percent indicated negative responses, *n* = 65 (Figure 1). Negative responses included “Ouch”, “Painful”, “Not in alignment”, and “First line looks more professional, bottom looks like a rush job”. A few patients noted that one line of incisions was straight, and one was not. Positive responses included “scary but considering if it is a spinal surgery—minimal”, “painful, but not too bad of a cut”, and “looks pretty minimal”. Patients were given a choice between which incision they preferred modified from Figure 2, excluding a single midline skin incision. The majority of patients chose traditional stab incisions (*n* = 41), followed by novel larger intersecting incisions (*n* = 37). The least popular incisions were the novel horizontal (*n* = 20) and the novel mini oblique (*n* = 5) incisions, Figure 2. When asked, “How much do you worry about how the incision on your back looks?” female patients worried more than male patients about how their incision looked. However, there was no statistically significant difference when evaluated by Mann–Whitney U two-tailed test (*p* value of 0.0418 via Mann–Whitney U one-tailed test and *p* value of 0.0836 via Mann–Whitney U two-tailed test). Patients less than or equal to 50 years of age worried more than patients over 51 years of age (*p* value of 0.0104 via Mann–Whitney U one-tailed test and *p* value of 0.0208 via Mann–Whitney U two-tailed test), which was statistically significant.

## 4. Discussion

MIS conjures up in both patient and surgeon the idea of a “band-aid” procedure. However, it does not consider the aesthetic layout of the incision. To date, lumbar paramedian incisions have been an undefined integral component of patient satisfaction with MIS. The genesis of this thought experiment fulfilled the doctoral research requirement of the first author. The first author, having 13 years of experience in spine surgery, found a disconnect between surgical exposure and patient satisfaction with postoperative scars. She asked herself the question, “Could MIS incisions larger than stabs be used to accomplish intraoperative surgical goals resulting in less operative time/radiation and improve patient satisfaction?” A comprehensive systematic review of patient-reported outcomes after lumbar surgery was conducted by Khan et al., 2015, which included 21 studies with 2734 patients [13]. The review found that patient satisfaction was generally high after lumbar surgery, with reported satisfaction rates ranging from 67% to 95%. Another systematic review, by Sivaganesan et al., 2020, evaluated patient satisfaction after lumbar spine surgery and found that overall patient satisfaction rates ranged from 64% to 96% [14]. The review also noted that patient satisfaction was higher among those who had minimally invasive surgery compared to those who had open surgery. In a study by Lubelski et al., 2014, patient satisfaction with lumbar microdiscectomy was evaluated using a questionnaire [15]. The study found that 91% of patients reported being satisfied with their surgery, and 92% reported an improvement in their symptoms. In a more recent study by Bouknaitir et al., 2021, patient satisfaction was evaluated after lumbar decompression surgery using a validated questionnaire [16]. The study found that 87% of patients reported being satisfied with their surgery, and 85% would choose to have the same surgery again. While these results affirm patient satisfaction regarding MIS surgery, they fail to address patient preferences with MIS skin incisions.

There has been a trend towards MIS due to the presumed advantages. These advantages include smaller incisions, reduced soft tissue disruption, reduced surgical blood loss, presumed less postoperative pain, reduced hospital stays, and faster return to employment. These achievements, if scientifically proven, come at a cost. McClelland and Goldstein performed a systematic review of 17 randomized controlled trials comparing MIS versus open surgery in 2017 [17]. They found that there was no evidence to support MIS over open surgery for lumbar or cervical disc herniation [17]. Moreover, they found that “in lumbar disc herniation, MIS was inferior in providing leg/low back pain relief, rehospitalization rates, quality of life improvement, and exposed the surgeon to greater than 10 times more radiation in return for shorter hospital stay and less surgical site infection” [17]. The effects of this increased radiation exposure on patients, surgeons, and staff have not been studied [17]. Many times, complications that might arise from performing decompression and fusion via an MIS approach could be avoided by converting to an open procedure [18]. A larger-than-stab incision may allow for near-open decompression and instrumentation while permitting less soft tissue disruption. The novel incisions presented here could allow for easier pedicle screw and rod in the case of scoliosis. This may be realized through open visualization and the ability to access two pedicles through one incision. The limitation of these novel incisions is that they are hypothetical. While the authors believe they are achievable, further feasibility study is needed, including surgeons’ opinions. 

Hamouda et al. present the results of a prospective randomized cohort study comparing patients who had a traditional vertical incision or a transverse incision [19]. The incisions were evaluated at the one-month postoperative visit using a color photograph of the incision shown to the patient. The conclusion was that a transverse incision is an acceptable alternative to traditional vertical incisions [19]. This study proposed that transverse incisions may result in better cosmesis as they follow Langer’s lines. “Transverse midline incisions have been used by other surgical specialties and provides a more cosmetically acceptable result,” explained Pencle et al. [19,20]. Deck and Kopriva found that vertical incisions may have a less favorable appearance when a delayed scar assessment scale is applied [21]. In post-surgical scar studies, patients have reported “feeling self-conscious, unattractive, or embarrassed about their scars” [22]. Others have correlated the perception of another’s pain according to the “threat value of pain” hypothesis [23], which activates the individual’s survival mechanisms, leading to withdrawal and avoidance as if exposed to real or perceived danger. This hypothesis stipulates that one’s ability to recognize peers in pain may help humans avoid danger and promote empathic behavior [24]. At least one study suggested that empathy-evoking stimuli produce automatic and controlled effects on both perceptual and motor processing, which could affect a patient’s clinical outcome assessment based on the perceived appearance of the incision as beautiful or not [24]. The impact of self-esteem and self-perceived body image on the acceptance of cosmetic surgery has also been well established [25].

There have been several postoperative scar measurement and evaluation tools created. Notably, Mundy et al. highlighted the deficiency in most of these scales in that they did not include an assessment of the scar appearance or patient satisfaction [26]. The authors suggest that a patient-reported-outcome (PRO)-centered assessment is needed [22,26,27]. This is supported in other related literature [22,26,27]. With over one million patients having surgery resulting in a scar, the results of a PRO tool could help guide surgical techniques in the future [22,26]. To take this a step further, this highlights the demand for patient input into their surgical incision before surgery, not after. The most noted scar scales in the medical literature include the following: the Vancouver Scar Scale; the Visual Analog Scale with scar ranking; the Patient and Observer Scar Assessment Scale (POSAS); the Manchester Scar Scale; the Stony Brook Scar Evaluation Scale; the Patient Scar Assessment Questionnaire (PSAQ); the Patient-Reported Impact of Scars Measure (PRISM); the Bock Quality of Life Questionnaire for Patients with Keloid and Hypertrophic Scarring (BOCK); and the SCAR-Q [22,23]. Of these scales, only the POSAS, Bock, PSAQ, and PRISM were PRO-oriented [24]. To address the limitations in the previously mentioned four scar scales, the SCAR-Q was developed and is being tested [22]. SCAR-Q “aims to measure outcomes specific to any type of scar” [22].

The authors desired to take advantage of the Typeform^TM^ platform ability to offer features such as logic jumps, which allow for dynamic branching of questions based on previous responses, and data piping, which allows for personalized questions based on previous responses. Had the authors not been restricted by the local IRB, a higher survey completion rate was expected. This is because of the Typeform™ survey methodology emphasizing a user-centric approach to survey design, aiming to provide an engaging and interactive experience for respondents. The total number of patients surveyed could have been larger and allowed for follow-up surveys comparing a single midline lumbar skin incision to lumbar paramedian MIS skin incisions. 

Our study showed that patients do reflect on the type of lumbar skin incisions that could be utilized intra-operatively. The majority of patients chose traditional stab incisions (*n* = 41), followed closely by novel larger intersecting incisions (*n* = 37), novel horizontal (*n* = 20), and mini oblique (*n* = 5). The first author prefers the novel mini oblique incision since it is transverse, larger than a stab incision, and would accommodate anatomic pedicle misalignment. However, the novel mini oblique incision was the least popular incision for patients. In addition, female patients and patients under 50 were more worried than male patients about how their incisions looked. While our study has limitations due to the small sample size induced by the local IRB and respondent bias, it suggests that patients’ input into potential surgical planning impacts patient satisfaction and should be taken into account. Finally, NSIMISS results suggest patients may find slightly larger novel skin incisions acceptable. Theoretically, use of a slightly larger skin incision offered by one of the novel incisions mentioned would permit enhanced surgical exposure and instrumentation. By converting from MIS stab incisions to novel MIS paramedian lumbar skin incisions, the number of incisions is reduced by half, which may improve patient satisfaction, requiring future study. Finally, utilizing slightly larger incisions may reduce intra-operative time and reduce radiation exposure to the patient and the surgeon and his team. Cadaveric human studies could further define the in situ feasibility of these novel incisions in regard to MIS exposure, decompression, interbody fixation, and instrumentation. Further study from spine surgeons is required to see how these novel incisions perform in surgery in relation to access and actual cosmetic result. These cosmetic results could be studied across larger demographics and ethnicities, focusing on post-surgical patients. 

Patients translate the length, location, and form of lumbar skin incision pain and recovery time needed to realize the desired benefit from the operation. While surgeons may find this irrational, patients’ perception of lumbar fusion surgery can vary depending on the individual patient’s pain tolerance, anxiety levels, and other factors. Although postoperative discomfort and pain, at least in the initial postoperative period, can be managed with local anesthesia or other pain relief measures, patients may also interpret the psychological burden from the operation differently if they experience swelling and bruising around the incision site with temporary numbness or tingling in the area. Using a PRO tool, each novel incision could be studied postoperatively and compared to stab incisions or to each other type of incision. 

One limitation of our study is the lack of functional outcome scores. These are typically aimed at evaluating patients’ physical function and ability to perform daily activities following surgery, as measured by objective criteria such as range of motion, strength, and mobility. The inability of these traditional functional outcome scores based on objective measures of physical function to account for subjective experiences, such as the physical appearance of the incision, prompted the authors to survey patients specifically regarding their lumbar MIS skin incision.

A patient’s perception of a skin incision can impact their recovery experience. It may influence their ability to participate fully in physical therapy or other aspects of their recovery program. For example, suppose a patient is experiencing significant pain or discomfort related to a cosmetically inferior or unappealing skin incision. In that case, they may be less likely to engage in exercises or activities necessary for optimal recovery. As such, healthcare providers must address any concerns or issues related to the skin incision to promote their patients’ best possible functional outcomes. The authors conclude that it is better to discuss the MIS skin incision pre-operatively to better meet the patients’ expectations for surgery, if those expectations do not hinder the surgeon. 

## 5. Conclusions

The clinical evidence regarding patient satisfaction with MIS lumbar surgery skin incisions is scant. Our study suggests that the type of lumbar spine skin incision is of importance to patients and that there may be a role for novel lumbar paramedian (MIS) skin incisions. Traditional patient-reported outcome measures measure functional improvements following spinal surgery but do not take patients’ perceptions of the skin incision into account. In an attempt to not conflate functional improvement measures with the assessment of the psychological impact of the skin incision on our patients, our patients were only surveyed regarding their possible skin incisions. Our research shows that managing patients’ expectations regarding their skin incisions is essential since individual expectations and experiences may vary. Surgeons must thoroughly discuss with patients their expectations, not just regarding potential functional outcome scenarios but also concerning the cosmetic appearance of the skin incisions. More reliable information about patients’ and surgeons’ perspectives regarding their scars should be collected. Using this information, surgeons might better plan minimally invasive, hybrid, or open techniques to improve patient satisfaction.

## Data Availability

The data presented in this study are available on request from the corresponding author.

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
