# Peer review of "Patient Perceptions of Paramedian Minimally Invasive Spine Skin Incisions"

_jpm, 2023, doi:10.3390/jpm13060878_

Round 1
Reviewer 1 Report
Although possibly interesting, the paper's topic is just showing the survey results. The results align with common sense and are not at all a surprise. However, I appreciate that the authors tried to evaluate such an aspect of spine surgical practice.
However:
1) Is a survey a prospective study? I do not believe so. Please correct it.
2) How many nonresponders did you have? In which percentage?
3) How much time from the surgery was the survey proposed? It may be possible that time influences the perception of the scar.
4) The authors stated that they have systematically reviewed the literature, but the process must be traced in the methods section. I think they should just say that they reviewed the literature.
Author Response
Reviewer #1
Although possibly interesting, the paper's topic is just showing the survey results. The results align with common sense and are not at all a surprise. However, I appreciate that the authors tried to evaluate such an aspect of spine surgical practice.
Response:
We thank the reviewer for his kind comments. The study findings appear obvious on their face. However, the existing literature on this subject is quite scarce. Also, common patient self-reported outcome measures do not account for the psychosocial impact the patient's perception of the skin incision may play on their overall reflection of the clinical success of their interventions. However, the skin incision is often the only aspect of their surgery the patients can see and show off to their family members and friends to communicate the severity of the operation and its associated health burden. Therefore, we conducted this survey to understand better how the lumbar spine's skin incisions may drive patients' perception of their clinical outcome.
Questions Reviewer #1:
- Is a survey a prospective study? I do not believe so. Please correct it.
Response: Yes, as stated in the method section this was a prospective study.
- How many nonresponders did you have? In which percentage?
Response: Only 3 patients did not complete the survey. Therefore, the response rate was 97.17%. We added the following sentence to the result section: “Hence, the response rate was 97.17%.”
- How much time from the surgery was the survey proposed? It may be possible that time influences the perception of the scar.
Response: We appreciate the reviewer pointing out this in sufficiency in our manuscript. The patients were surveyed three to six months postoperatively. We added the following sentence to the method section under study design: “Patients were surveyed three to six months postoperatively.”
- The authors stated that they have systematically reviewed the literature, but the process must be traced in the methods section. I think they should just say that they reviewed the literature.
Response: We thank reviewer for his correction. We deleted any reference from the paper that would suggest we performed a systematic literature review. It simply states that we reviewed the available literature on the skin incision topic.
Reviewer 2 Report
It may be an interesting report on paramedian skin incisions in minimally invasive spine surgery, but this reviewer has doubts about patient satisfaction. With this method, the authors were only asking about the skin incision preferences of the patients who participated in this study. It is far from assessing patient satisfaction with minimally invasive spine surgery.
Author Response
We thank the reviewer for the kind comments and understand the concern. We intentionally did not question patients about the functional outcome measured with traditional patient-reported outcome measures. We did not want to conflate the functional outcome question with the straightforward questions regarding the skin incisions. We hope this reviewer finds the explanation acceptable.
Reviewer 3 Report
1. In Abstract: In sentence “Patient < or equal to 50 years”, author should used term “less than or equal to 50 years” instead of using symbol “<”.
2. In figure 2: There is not description or detail of Figure 2C. In addition, figure 2B, picture of skin incision on right side is not mini-open, it is conventional midline incision. Could you correct label and description of this figure?
3. In Materials and Methods: Author should give more detail about MIS surgery (Fusion or non-fusion surgery) (LLIF with Percutaneous screws or MIS TLIF). Because, in Figure 1, the surgical wound is representing only percutaneous screw insertion.
4. In Materials and Methods: Author should calculate sample size base on previous study. Because there were many previous studies about satisfaction of patient after spine surgery.
The quality of English is very good.
Author Response
Questions Reviewer #3
- In Abstract: In sentence “Patient < or equal to 50 years”, author should used term “less than or equal to 50 years” instead of using symbol “<”.
Response: We complied with the reviewers request and changed the sentence to read: ” Patients less or equal to 50 years of age…”
- In figure 2: There is not description or detail of Figure 2C. In addition, figure 2B, picture of skin incision on right side is not mini-open, it is conventional midline incision. Could you correct label and description of this figure?
Response: At the reviewers request, we changed the legend of figure 2 and added labels d and e to avoid confusion. The legend for figure 2 now reads: “Figure 2. a) Traditional stab incision and b) novel mini-open horizontal incisions compared to c) traditional single midline, d) oblique intersecting and e) vertical mini-open incisions.”
- In Materials and Methods: Author should give more detail about MIS surgery (Fusion or non-fusion surgery) (LLIF with Percutaneous screws or MIS TLIF). Because, in Figure 1, the surgical wound is representing only percutaneous screw insertion.
Response: We added the requested additioinal information by adding to the method section under 2.1. the following sentence: “Patients were surveyed three to six months postoperatively after having undergone minimally invasive transforaminal lumbar interbody fusion with interbody fusion cages, and pedicle screws through mini-open skin incisions.”
- In Materials and Methods: Author should calculate sample size base on previous study. Because there were many previous studies about satisfaction of patient after spine surgery.
Response: Our patients were only surveyed regarding their perceptions of various forms of minimally invasive lumbar skin incisions. We did not test any treatment agaiinist another. Our study was an observational cohort study without a control group with simple descriptive statistics on the survey responses. Therefore, a sample size calculation was not feasible.
Round 2
Reviewer 2 Report
Patient's satisfaction with surgical procedures involved skin incision is not different from patient preference and expectation for surgery. The title should be changed.
Author Response
We appreciate this reviewers recommendation and changed the title to:
"Patient Perceptions of Paramedian Minimally Invasive Spine Skin Incisions"
Reviewer 3 Report
Dear, authors
About picture of skin incision for MIS TLIF in Figure 2, normally on side in which surgeon performed interbody fusion should be bigger than the other side in which performed only percutaneous screw. Could author explained about this?
The quality of English language is very good
Author Response
We understand the reviewer's question. To clarify what surgery was offered to patients we added the word bilateral to the following sentence to section 2.1. Study Design. This section now reads: "Patients asked to contemplate minimally invasive transforaminal lumbar interbody fusion with bilateral interbody fusion cages, and pedicle screws through mini-open skin incisions were surveyed." Thus, we did imply to patients that they would undergo a bilateral pedicle screw, rod, and cage implantation. Hence, the skin incisions needed to be the same size.